

# Methods for validation of random uncertainty estimates and their applications to ozone profiles from limb-viewing satellite instruments

Viktoria F. Sofieva[1], Alexandra Laeng[2], Thomas von Clarmann[2†], Gabriele Stiller[2], Michael Kiefer[2], Johanna Tamminen[1], Alexey Rozanov[3], Carlo Arosio[3], Nathaniel Livesey[4], Robert Damadeo[5], Patrick Sheese[6], Kaley A. Walker[6], Doug Degenstein[7], Daniel Zawada[7], Natalya A. Kramarova[8], Arno Keppens[9]

[1] Finnish Meteorological Institute, Helsinki, Finland
[2] Karlsruhe Institute of Technology, Institute of Meteorology and Climate Research - Atmospheric Trace Gases and Remote Sensing, Karlsruhe, Germany
[3] University of Bremen, Bremen, Germany
[4] Jet Propulsion Laboratory, California Institute of Technology, Pasadena, California, USA
[5] NASA Langley Research Center, Hampton, VA, USA
[6] Department of Physics, University of Toronto, Toronto, Canada
[7] Institute of Space and Atmospheric Studies, University of Saskatchewan, Saskatoon, Canada
[8] NASA Goddard Space Flight Center, Greenbelt, Maryland, USA
[9] Royal Belgian Institute for Space Aeronomy (BIRA-IASB), Brussels, Belgium

[†] deceased

*Correspondence to*: V.F. Sofieva (viktoria.sofieva@fmi.fi)

## Abstract

For satellite measurements of atmospheric composition, the random uncertainty estimates provided by retrieval algorithms might be imperfect due to various approximations used in the retrievals or presence of unknown error sources. This paper presents an overview of the methods used for validation of random uncertainty estimates. All methods discussed in this study are categorized, and assumptions and limitations of each method are discussed. This overview evaluates these methods in application to ozone profile measurements data from limb and occultation satellite instruments and provides practical illustrations of random uncertainty validation.

## 1    Introduction

In nearly all data analyses, such as data comparisons, aggregating/combining/merging data, data assimilation etc., information about data uncertainty is needed. Such characterization of uncertainty would ideally include both systematic and random components, as well as spatio-temporal resolution of the data, as discussed in von Clarmann et al. (2020). Validation of uncertainty estimates is needed, especially if the measurement uncertainty cannot be fully characterized or is based on assumptions. This is typical for remote-sensing measurements, which use retrievals of atmospheric parameters that solve inverse problems. Random uncertainty of the remote sensing measurements is usually estimated via propagation of instrumental noise and other random uncertainties through the inversion algorithm. These estimates, which are sometimes referred to as "ex-ante" errors (von Clarmann, 2006) (other terms are "prognostic", "predicted", "inductive", or "bottom-up"), can be imperfect due to various approximations used in retrievals, or due to the presence of random components in parameter uncertainties.

The aim of this paper is to provide an overview of the methods for validation of random error component. We extend the overview of such methods presented in the introduction of Sofieva et al. (2014) and illustrate them using ozone profile



retrievals from Earth-orbiting limb and occultation instruments. In our paper, we discuss the applicability and limitations of each method, with the focus on ozone profile retrievals from satellite measurements.

This paper contributes to the APARC (Atmospheric Processes and their Role in Climate) activity TUNER (Towards UNified Error Reporting) https://www.aparc-climate.org/activities/tuner/.

5 **2    Data**

To illustrate the methods employed for validation of random uncertainties, we use ozone profiles retrieved from several limb and occultation measurements. The summary of the datasets is collected in Table 1, and the principles of uncertainty estimates are described below.

10 **Table 1. Information about the ozone profile datasets used in the paper.**

| Instrument/ satellite/processor | Principle of retrieval appproach/uncertainty estimation and references | Time period | Vertical resolution | Estimated random uncertainty in the stratosphere | Profiles per day |
|---|---|---|---|---|---|
| SAGE II/ ERBS NASA v7.0 | Error propagation (Damadeo et al., 2013) | Oct 1984 – Aug 2005 | ~1 km | 0.5–5% | 14–30 |
| OSIRIS/ Odin USask v7.2 | Error propagation (Bourassa et al., 2018) | Nov 2011 – present | 2–3 km | 2-10% | ~250 |
| GOMOS/ Envisat ALGOM2s v1.0 | Error propagation of instrumental noise and residual scintillation error (Kyrölä et al., 2010; Sofieva et al., 2017) | Aug 2002 – Aug 2011 | 2–3 km | 0.5–5 % | ~110 |
| MIPAS/ Envisat KIT/IAA V8 | Error propagation (von Clarmann et al., 2009; Kiefer et al., 2023) | Jan 2005 – Apr 2012 | 3–5 km | 1–4% | ~1000 |
| SCIAMACHY/ Envisat UBr v3.5 | Error propagation for retrieval noise, parameter errors are estimated for selected representative scenarios using the Monte-Carlo approach (Jia et al., 2015; Rahpoe et al., 2013) | Aug 2003 – Apr 2012 | 3–3.5 km | 1–7% | ~1300 |
| ACE-FTS/ SCISAT V5.2 | Least-squares statistical fitting errors (Boone et al., 2005; Sheese et al., 2022) | Feb 2004 – present | ~3 km | 1–4% | ~30 |
| MLS/Aura NASA v.5 | Error propagation (Livesey et al., 2006; Read et al., 2006) | 2004- Present | ~ 3 km | 2-5 % | ~3500 |
| OMPS-LP/ Suomi NPP USask 2D v1.3.0 | Error propagation of measurements uncertainty (Zawada et al., 2018) | Apr 2012 – present | ~2 km | 2–10% | ~2000 |
| OMPS-LP/ Suomi NPP UBr v4.1 | Error propagation for retrieval noise, parameter errors are estimated for selected representative scenarios using the Monte-Carlo approach (Arosio et al., 2022) | Apr 2012 – present | ~2-3 km | 3-5% | ~2000 |
| OMPS-LP/Suomi NPP NASA v2.6 | Error propagation (Kramarova et al., 2024) | April 2012- present | 1.9 – 2.5 km | 3-5% | ~2000 |
| SAGE III /ISS NASA AO3 v5.3 | Error propagation (Wang et al., 2020) | 2017 – present | ~1 km | 2–4% | ~30 |



### 2.1    Atmospheric Chemistry Experiment -Fourier Transform Spectrometer (ACE-FTS)

In retrievals from ACE-FTS, estimated random uncertainties of ozone profiles are the fitting errors from the least-squares inversion process (Boone et al., 2005, Sheese et al., 2022). The mean relative random uncertainties are estimated to be lower than 3% between 12 and 62 km and typically less than 2% around 30–35 km. Relative uncertainties are slightly higher in polar

regions. The ACE-FTS ozone uncertainty estimates slightly grow with time, going from ~1.7% in the middle stratosphere in the beginning of the mission to ~2.0% in the recent period. The vertical resolution of ACE-FTS ozone profiles is estimated to be ~ 3 km.

### 2.2    Global Ozone Monitoring by Occultation of Stars (GOMOS)

In this paper we use the GOMOS ozone profiles processed with ALGOM2s v.1 Scientific Processor (Sofieva et al., 2017). The

error propagation scheme is similar to that used in GOMOS IPF v.6 processor (Kyrölä et al., 2010; Tamminen et al., 2010), as the ALGOM2s ozone profiles are identical to those of IPF v.6 in the stratosphere and differ only in the UTLS.  The error estimates (square roots of the diagonal elements of the covariance matrix) are provided in the Level 2 data. The covariance matrix of retrieved profile uncertainties is obtained via Gaussian error propagation through the GOMOS inversion, see Tamminen et al. (2010) for details. Both noise and the dominating random modelling error (due to scintillations) are taken into

account in GOMOS inversion. Thus, error estimates provided in Level 2 files represent the total random uncertainty estimates (if neglecting the random part of the parameter errors, which are expected to be minor compared to the abovementioned random error sources).

The random uncertainties of GOMOS ozone profiles depend on stellar brightness, spectral class and obliquity of occultation. They are typically in the range from 0.5% to 5% in the stratosphere. Examples of typical uncertainties of GOMOS ozone

profiles can be found in Tamminen et al. (2010). An extensive validation of GOMOS random uncertainty estimates is performed and reported in Sofieva et al. (2014). It was shown that GOMOS random uncertainty estimates are realistic for not-dim stars. Due to instrument ageing, GOMOS random uncertainty estimate grow with time, especially for dim stars (Tamminen et al., 2010).

### 2.3    Michelson Interferometer for Passive Atmospheric Sounding (MIPAS)

The random errors provided with version 8 IMK MIPAS data consist of the propagated covariance of the spectra, and further uncertainties of parameters used in the retrievals (like uncertainty of the temperature or the line-of-sight pointing) which are of random nature. A detailed description of how these random errors were calculated is provided by von Clarmann et al., (2022) and Kiefer et al. (2023). According to Kiefer et al. (2023, Supplement), the total random error is up to about a factor of 2 to 3 (even >3 for unfavorable conditions like polar winter) higher than the pure measurement noise error in the lower part of

the stratosphere (up to about 30 km), and by a factor of 1.1 to 1.4 higher in the upper part of the stratosphere. For illustrations in our paper, if not specified explicitly, we use the total random uncertainty.

### 2.4    Microwave Limb Sounder (MLS)

This paper uses the Aura MLS "Version 5" dataset (Livesey et al., 2024), retrieved using the same tomographic retrieval algorithm employed for all previous MLS data versions (Livesey et al., 2006; Read et al., 2006; Schwartz et al., 2006). The

Level 2 data products consist of vertical profiles spaced 1.5º along the orbital track, with pressure as the vertical coordinate. Each profile is accompanied by a separate profile reporting the estimated precision uncertainty (i.e., random noise) in the profile. This is based on the square root of the diagonal of the solution covariance matrix from the optimal estimation-based retrieval and thus overestimates the scatter in the geophysical products in cases where the a priori information and other regularization constraints contribute significantly to the results (typically at the upper end of the useful vertical range of each

product). Systematic errors due to uncertainties in instrument calibration, spectroscopy, and other parameters have been

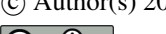


quantified through multiple perturbation studies detailed in Read et al. (2007), and are documented for each version, along with averaging kernels and rules for data use and screening, in a data quality document (Livesey et al., 2024 for version 5).

### 2.5     Ozone Mapper and Profile Suite - Limb Profiler (OMPS-LP)

For illustrations in this paper, we use three ozone profile datasets from OMPS-LP instrument: OMPS-LP USask 2D v1.3.0 processed at University of Saskatchewan, OMPS-LP UBr v.4.1 processed at University Bremen, and the NASA processor v. 2.6

The OMPS-LP USask two-dimensional retrieval process uses Gaussian error propagation to estimate the covariance of the retrieved solution due to measurement noise (ignoring the smoothing error). The reported precision is the square root of the

diagonal elements of the converged solution covariance matrix. The measurement noise is assumed to be a constant 1% at all altitudes and wavelengths.

OMPS-LP UBr v.4.1 Level 2 data provide covariance matrices for the retrieved profiles which are obtained by the propagation of the measurement noise errors. The latter are estimated from spectral fit residuals obtained during the pre-processing step. Contributions of the parameter errors are estimated using the Monte-Carlo approach for a set of representative observations

and reported by (Arosio et al. 2022) along with the total error budget estimations.

The NASA OMPS-LP v2.6 algorithm (Kramarova et al., 2024; Rault and Loughman, 2013) retrieves ozone profiles by employing the second order Tikhonov regularization method. The estimated precision for each profile retrieval is calculated using the square roots of diagonal elements of the solution covariance matrix. Systematic errors related to uncertainties in altitude registration, some algorithmic parameters, and apriori profiles have been evaluated and reported by (Kramarova et

al., 2024; Moy et al., 2017).

### 2.6     Optical and Spectroscopic Remote Imaging System (OSIRIS)

OSIRIS V7.2 data uses standard Gaussian error propagation to estimate the uncertainties in the retrieved ozone profiles. The covariance matrix is calculated through propagation of the measurement noise that is estimated through counting statistics. The reported precision is the square root of the diagonal elements of the covariance matrix.

### 25   2.7     Stratospheric Aerosol and Gas Experiment (SAGE)

The retrieval algorithms for both SAGE II v7.0 (Damadeo et al., 2003) and SAGE III/ISS v5.3 (see SAGE III ATBD (Wofsy et al., 2002) and Wang et al., 2020) are similar. The uncertainties are computed from the statistical distribution of observations in the L1 transmission data and then propagated through the retrieval algorithm via Gaussian error propagation into the uncertainties for the L2 profiles of ozone and other species. Typical retrieval noise values are within 1 % between 18 and 52

km.

### 2.8     Scanning Imaging Absorption Spectrometer for Atmospheric Cartography (SCIAMACHY)

SCIAMACHY V3.5 Level 2 data provide covariance matrices for the retrieved profiles which are obtained by the propagation of the measurement noise errors. The latter are estimated from spectral fit residuals obtained during the pre-processing step. Contributions of the parameter errors are estimated using the Monte-Carlo approach for a set of representative observations.

Rahpoe et al. (2013) described the method to estimate the parameter errors and reported resulting values along with the total error budget estimations for the precursor retrieval version (V2.5). The results for V3.5 are expected to be similar.



## 3    Methods for validation of random uncertainty estimates

### 3.1    Specifics of satellite measurements of atmospheric composition

In remote sensing, retrieved parameters result from solving the inverse problem. The reported random uncertainties are usually estimated via propagation of instrumental noise and other random errors through the inversion algorithm. Error

estimates might be wrong if the source uncertainties used for propagation are not known well enough. If some of the error sources are not characterized and the corresponding uncertainties are not considered, the reported uncertainty is underestimated.

The normalized $\chi^2$ statistics, $\chi^2_{norm}$, is commonly used for assessing the adequacy of the theoretical description of measurements (forward model) and as an indication of the correctness of random uncertainty estimates. $\chi^2_{norm}$ is usually

evaluated as:

$$\chi^2_{norm} = \frac{1}{N-p} \left( \boldsymbol{y} - \boldsymbol{y}_{mod} \right)^T \mathbf{S}^{-1}_{y,random} \left( \boldsymbol{y} - \boldsymbol{y}_{mod} \right), \tag{1}$$

where $\boldsymbol{y}$ is the vector of observed parameters (e.g., spectrally resolved radiance values, transmittances), $\boldsymbol{y}_{mod}$ is the vector of modelled (theoretical) measurements, $\mathbf{S}_{y,\,random}$ is the covariance matrix of random measurement errors, $N$ is the number of measurements and $p$ is the number of retrieved parameters (e.g., Bevington and Robinson, 2003; Taylor, 1997). $\chi^2_{norm}$ is also

called "$\chi^2$ per degree of freedom". If the theoretical model describes the experimental data correctly and the measurement errors are properly defined, $\chi^2_{norm} \approx 1$, ideally. Very large $\chi^2_{norm}$ values indicate underestimated random uncertainties, while $\chi^2_{norm}$ smaller than 1 imply that they are overestimated (Bevington and Robinson, 2003). This simple analysis of $\chi^2_{norm}$ has helped to discover a missing random error component in early GOMOS retrievals, which was due to uncorrected residual scintillation, and to parameterize it in later processing versions (Sofieva et al., 2010; Tamminen et al., 2010). In the first step

of the GOMOS retrievals – the spectral inversion (Kyrölä et al., 2010) - $\chi^2_{norm}$ is evaluated at each tangent altitude independently; $\boldsymbol{y}$ and $\boldsymbol{y}_{mod}$ in the GOMOS case are measured and modelled transmittance spectra, $N$ is number of spectral pixels (maximum 1416) and $p$ is the number of fitted parameters (slant column densities for ozone, $NO_2$, $NO_3$, and aerosol parameters). Figure 1 compares $\chi^2_{norm}$ in the GOMOS retrievals in the set of oblique occultations from the brightest star Sirius when residual scintillation errors are ignored (blue) or considered (red).



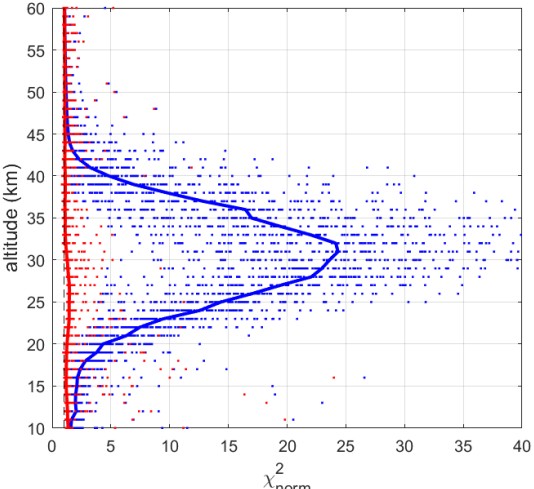

**Figure 1. Adapted from (Sofieva et al., 2010):** $\chi^2_{norm}$ **in GOMOS retrievals from a set of oblique occultations of Sirius (August 2003, 66°S, obliquity angle is ~25°). Blue: random error due to residual scintillation is ignored, red: modelling errors are taken into account in the retrievals. Dots: values in individual occultations, bold lines indicate median values for the sets. The black**
**dashed line indicates** $\chi^2_{norm} = 1$

A similar analyses of $\chi^2_{norm}$ applied to GOMOS IPF v6 data has spotted $\chi^2_{norm}$ <1 at upper altitudes in case of dim stars, and a more detailed analysis identified the reason and the influence on uncertainty estimates (Sofieva et al., 2014).

In some retrievals, the uncertainty estimate is derived from the fit residuals, as it is done for SCIAMACHY and OMPS-LP

UBr ozone retrievals (e.g., Arosio et al., 2022). Such an approach forces $\chi^2_{norm}$ to be close to 1 and can provide an estimate the uncertainties of measurements in the case one does not trust the information on measurement uncertainty contained in the Level-1 data.

$\chi^2_{norm}$ is a statistical and integral characteristic that indicates not only correctness of the random uncertainty estimates but also consistency of measurements with the forward model used for inversion of atmospheric parameters, which

is itself often incomplete, or reflects inaccurate knowledge of instrument calibration and/or spectroscopic parameters. In some cases, a priori/regularization terms are included into $\chi^2_{norm}$ too. Sometimes, this metric is used to determine how strong the regularization should be. Therefore, analyses of $\chi^2_{norm}$ at the measurement level are useful, but they cannot substitute validation of uncertainty estimates of the retrieved ozone profiles.

The retrieved ozone profiles are characterized not only by uncertainty estimates but also by vertical resolution and, in the case

of tomographic retrievals, by horizontal resolution. This means that, for such retrievals, the impacts of radiance noise on adjacent data points are not independent, and the proper characterization of associated uncertainties is obtained using their covariance matrix.

If the retrieval is performed with the Bayesian maximum a posteriori estimates (Rodgers, 2000; von Clarmann, 2006), a data correlation can also arise due to the usage of a priori information. These aspects should be taken into account when validating

uncertainties.



### 3.2 General strategy

In the laboratory, experimental precision estimates can be obtained using repeated measurements under the same conditions: the sample variance $s^2 = \mathrm{var}(x) = \left\langle (x - \langle x \rangle)^2 \right\rangle$ (angular brackets denote the mean hereafter) approaches the variance of random error distribution $\sigma^2$ (i.e., squared precision) when the size of sample $N$ tends to infinity. For different samples of size

$N$, the values of sample variance will vary due to different random error realization. An ensemble of sample variances is a random variable with a distribution depending on noise variance $\sigma^2$ and $N$. The quantity $\dfrac{(N-1)s^2}{\sigma^2}$ has a $\chi^2$ distribution with

$N-1$ degrees of freedom, $\chi^2_{N-1}$ ( e.g., Bevington and Robinson, 2003; Taylor, 1997). For large $N$, $\chi^2_{N-1}$ distribution can be

approximated by a Gaussian distribution with variance $2N$ thus $\mathrm{var}\left( \dfrac{(N-1)s^2}{\sigma^2} \right) = 2N$ . This gives the uncertainty of the

experimentally estimated random error

$$\mathrm{var}(s^2) \approx \sigma^4 \frac{2}{N} .\tag{2}$$

In contrast with many laboratory experiments, geophysical observation conditions cannot be kept exactly constant for atmospheric measurements. Therefore, the sample variance contains a contribution from the natural variability $\sigma^2_{nat}$ :

$$s^2 \approx \sigma^2 + \sigma^2_{nat} .\tag{3}$$

For validation of uncertainty estimates, $\sigma^2_{nat}$ should be minimized by selecting collocated measurements or it should be

estimated from independent sources (for example, from a chemistry-transport model, CTM).

Approaches for validation of error estimates usually rely on the variance of the difference, $s^2_{12} = \mathrm{var}(x_1 - x_2),$ in a set of collocated measurements $x_1$ and $x_2$:

$$s^2_{12} = \sigma^2_{0,nat} + \sigma^2_1 + \sigma^2_2 .\tag{4}$$

In Eq.(4), $\sigma^2_{0,nat}$ stands for the natural variability within a space-time collocation window (note that $\sigma^2_{0,nat}$ , which represent

the mismatch uncertainty, is different from $\sigma^2_{nat}$ (natural variability in a certain  location) in Eq.(3)).

It is important to note that for the vertically resolved ozone profile data involved, calculating differences and combining data require harmonization of data representations in terms of physical quantities and vertical sampling at least. As the satellite data result from a retrieval process, knowledge of prior information and averaging kernel matrices in principle allows retrieval differences to be accounted for as well. Keppens et al. (2019) provide an overview of harmonization operations for atmospheric

profile observations is provided, covering vertical representation matching, vertical smoothing matching, and retrieval matching (essentially the prior information contributions). The effect of these manipulations on the information content and uncertainty budget of the original data is extensively discussed in that work and will not be repeated here.  In the following, we assume that all profiles are presented in a similar vertical resolution. For the illustrations in this paper, the harmonization of the vertical resolution is not needed, as the satellite limb profiles have similar vertical resolution, see Table 1.


We divide the methods for random uncertainty validation into two groups depending on what kind of data are used: (1) from the same instrument and (2) from different instruments.



### 3.3    Using collocated measurements from the same instrument

For perfectly collocated measurements ($\sigma_{0,nat}^2 \approx 0$) from the same instrument with the same precisions $\sigma_1 = \sigma_2 = \sigma$, Eq.(4) is reduced to $s_{12}^2 \approx 2\sigma^2$, thus allowing validation of the uncertainty estimate $\hat{\sigma}^2 = s_{12}^2/2$. In this estimate, random errors in $x_1$ and $x_2$ are assumed to be uncorrelated. This uncertainty validation method was realized, for example, for closely

collocated MIPAS ozone profiles (Piccolo and Dudhia, 2007) and OSIRIS ozone measurements (Bourassa et al., 2012). The uncertainty of this experimental precision estimate is defined by the uncertainty of sample variance $s_{12}^2$.

There are several limitations associated with this method. First, natural variability $\sigma_{0,nat}^2$ is not necessarily small, even with a tight spatio-temporal window. In such cases $s_{12}^2$ will be larger than a combined uncertainty $2\sigma^2$, thus the estimate $\hat{\sigma}^2 = s_{12}^2/2$ will be biased high. Second, the number of self-collocated measurements for limb satellites is limited (self-collocated

measurements are usually found around the Poles, while at other latitudes a larger temporal separation has to be accepted). In addition, the number of self-collocated measurements for instruments with coarse sampling (stellar and solar occultation) is relatively low. For example, $\sim 200$ collocated occultations (with spatial separation $\Delta r$ less than 300 km and temporal separation $\Delta t$ less than 3h) per year of the star S30 (notation in the GOMOS catalogue) can be found for GOMOS; all located near the North Pole in winter. For ACE-FTS, fewer than 100 self-collocations per year with the criteria

$\Delta t = 3\text{h}, \ \Delta r = 300 \ \text{km}$ are found, and ~400 self-collocated measurements per year can be found with the collocation criteria $\Delta t = 5\text{h}, \ \Delta r = 500 \ \text{km}$. For SAGE II and SAGE III/ISS, there are no self-collocated measurements with the abovementioned collocation criteria.

Provided many collocated measurements from the same instrument are available (self-collocations), the precision of the dataset can also be estimated by computing a so-called structure function $D(\boldsymbol{\rho})$ (e.g., Tatarskii, 1961), or the RMS

difference of the field $f(\mathbf{r})$ as a function of increasing separation in time and in space:

$$D(\boldsymbol{\rho}) = D(\mathbf{r}_1 - \mathbf{r}_2) = \frac{1}{2}\left\langle \left[ f(\mathbf{r}_1) - f(\mathbf{r}_2) \right]^2 \right\rangle \tag{5}$$

where $\mathbf{r}_1$ and $\mathbf{r}_2$ are two locations and a vector $\boldsymbol{\rho} = \mathbf{r}_1 - \mathbf{r}_2$ is their spatio-temporal separation. In geostatistics, $D$ is called the variogram (Cressie, 1993; Matheron, 1963; Wackernagel, 2003). When using experimental (noisy) data for evaluation of the variogram/structure function, the difference of an atmospheric parameter in two locations is defined not only by the natural

variability of this atmospheric parameter, but also by uncertainty of the measurements. Therefore, with the spatio-temporal separation $\rho \to 0$, $D(\boldsymbol{\rho})$ tends toward the random uncertainty variance $\sigma_{noise}^2$ (the offset at zero is called "nugget" in geostatistics). Since self-collocated measurements are from the same instrument, no biases between them are expected. Figure 2 illustrates the structure function method, which is discussed in details in Sofieva et al. (2021) and applied to TROPOMI total ozone measurements.






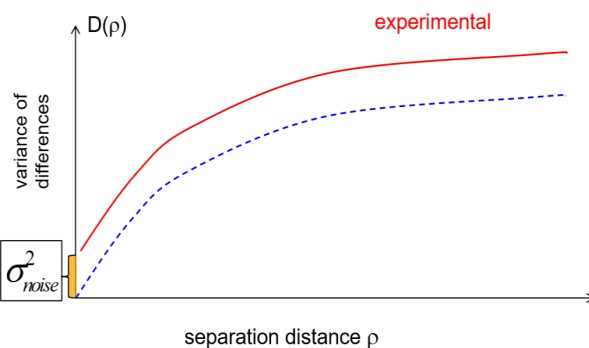

**Figure 2. Reproduced from (Sofieva et al., 2021):** *The schematic representation of the structure function estimated from noisy measurements, ρ denotes spatio-temporal separation.*

For ozone profiles from limb instruments, the structure function method is difficult to apply, as it requires a substantial number of measurements with close separation. An analogous method - evaluation of the one-dimensional structure function in polar regions (with transformation of temporal mismatch to spatial separation using the ECMWF wind field) - has been applied for validation of random uncertainty estimates of the MIPAS and GOMOS ozone profiles (Laeng et al., 2015; Laeng and Von Clarmann, 2021; Sofieva et al., 2014).

Figure 3 illustrates the application of the structure function method to MIPAS v8 ozone profiles, which have a detailed error characterization. With decreasing separation distance between measurements, ex-post uncertainties $S_{12}/\sqrt{2}$ approach to a curve, which is between ex-ante estimates for total random error (thick red curves) and for instrumental noise (thick magenta curves) .

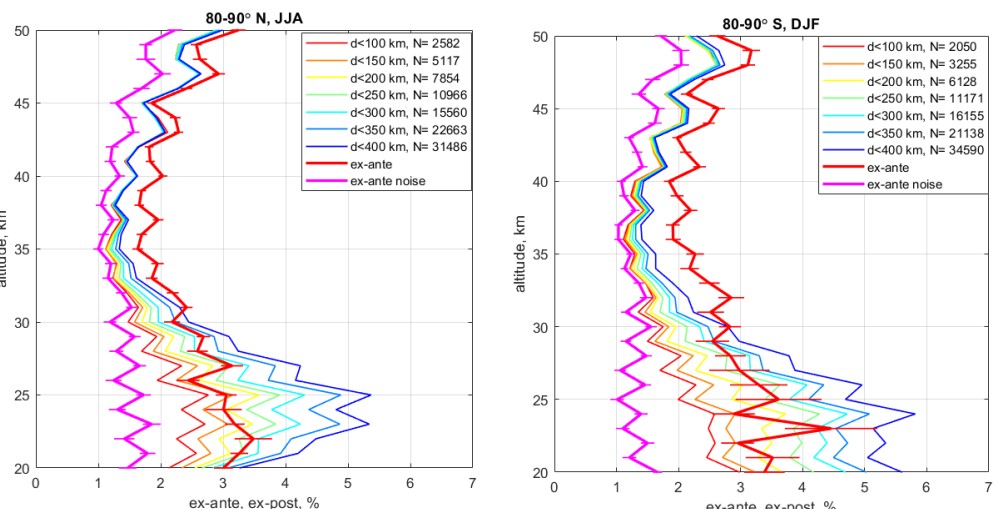

**Figure 3. Colored thin vurves: experimental precision estimates** $S_{12}/\sqrt{2}$ **for different separation distances; thick curves with errorbars: the mean ex-ante uncertainty estimate its standard deviation for propagated instrumental noise (magenta) and full random uncertainty (red); MIPAS self-collocations close to the North and South Poles in 2005–20011 during local summer are used.**



### 3.4 Using measurements from different instruments

#### 3.4.1 Method of Fioletov et al. (2006)

Fioletov et al (2006) have proposed estimating simultaneously the random data uncertainties and natural variability from sample variances of two perfectly collocated datasets and the variance of their difference. We reproduce the formulae here, as we assess the application of this method. The Fioletov method relies on sample variances $s_i^2$ of the collocated data:

$$s_i^2 = \sigma_{nat}^2 + \sigma_i^2, \quad i = 1, 2 \tag{6}$$

and the variance of their difference (Eq. (4)), which is reduced to:

$$s_{12}^2 = \sigma_1^2 + \sigma_2^2, \tag{7}$$

by assuming $\sigma_{0,nat}^2 \approx 0$. It is also assumed that the bias between datasets is the same for the selected sample.

In (6) and (7), $\sigma_{nat}^2$ is natural variability and $\sigma_i^2$ are measurement precisions. Solving (6) and (7) for $\sigma_{nat}^2$, $\sigma_1^2$ and $\sigma_2^2$, we get their experimental estimates based on sample variance:

$$\hat{\sigma}_{nat}^2 = 0.5\left(s_1^2 + s_2^2 - s_{12}^2\right)$$
$$\hat{\sigma}_1^2 = 0.5\left(s_1^2 - s_2^2 + s_{12}^2\right) \tag{8}$$
$$\hat{\sigma}_2^2 = 0.5\left(s_2^2 - s_1^2 + s_{12}^2\right)$$

The uncertainty of the natural variability and precision estimates given by (8) depend on uncertainty of sample variances, which depend, in turn, on sample variances themselves and the number of measurements. The estimates are thus only as accurate as the least accurate of these parameters. In approximation of large samples (when uncertainty of the sample variance can be approximated by Eq.(2)), the variance of the estimates (8) can be expressed in terms of „true" natural variability and precision variances $\sigma_{nat}^2$, $\sigma_1^2$ and $\sigma_2^2$ as (using Eqs (2, 6-8)):

$$\text{var}(\hat{\sigma}_1^2) = \text{var}(\hat{\sigma}_2^2) = \text{var}(\hat{\sigma}_{nat}^2) = \frac{1}{2N}\left(\left(\sigma_{nat}^2 + \sigma_1^2\right)^2 + \left(\sigma_{nat}^2 + \sigma_2^2\right)^2 + \left(\sigma_1^2 + \sigma_2^2\right)^2\right) \tag{9}$$

with the following simple estimates for upper and lower limits (after opening brackets in Eq.(9)):

$$\frac{1}{N}\left(\sigma_{nat}^4 + \sigma_1^4 + \sigma_2^4\right) < \text{var}(\hat{\sigma}_{1,2,nat}^2) < \frac{1}{N}\left(\sigma_{nat}^2 + \sigma_1^2 + \sigma_2^2\right)^2. \tag{10}$$

Since the precision estimates by the Fioletov method are linear combinations of three sample variances, they can have large uncertainty if one of the sample variances is large and/or the number of collocated measurements is limited. Not for all combinations of limb instruments perfectly collocated measurements can be found (especially for instruments with not dense sampling). In practice, satellite measurements separated by a few hundreds of kilometers and a few hours are considered collocated. The natural variability within the space-time collocation window is small but not zero. This results in additional difficulties in the application of this method. Note that the estimates from Eq.(8) do not ensure positivity of $\sigma_1^2$ and $\sigma_2^2$. Negative solutions can be within uncertainty intervals; their appearance can be caused either by insufficient amount of data or by the unaccounted natural variability within the collocation window.

For illustration, we applied this method to MIPAS and SCIAMACHY measurements in 2007. Collocated profiles with time separation less than 5 h, spatial distance less than 400 km and latitude difference less than 2° were selected in the tropics (20°S–20°N), with13785 such profile pairs found. The left panel of Figure 4 shows sample standard deviations $s_1$ and $s_2$ of MIPAS and SCIAMACHY profiles, respectively, and the standard deviation of differences $s_{12}$. The right panel shows the a-





posteriori (ex-post) estimates of random uncertainties and natural variability from Eq.(8) with uncertainties therein given by Eq.(9). The estimates of random errors reported by the retrieval algorithms ("ex-ante" in terminology of (von Clarmann et al., 2020)) are also shown in right panels of Figure 4 by dashed lines. Negative estimates of $\sigma_1^2$ and $\sigma_2^2$ are ignored. All computations are performed in absolute units, but the estimates are plotted as a percentage for clarity. We observe that the ex-ante and ex-post uncertainties of MIPAS ozone profiles are very close to each other. For SCIAMACHY, Fioletov's method suggests a larger uncertainty estimates at altitudes 25–37 km than reported in the retrievals.

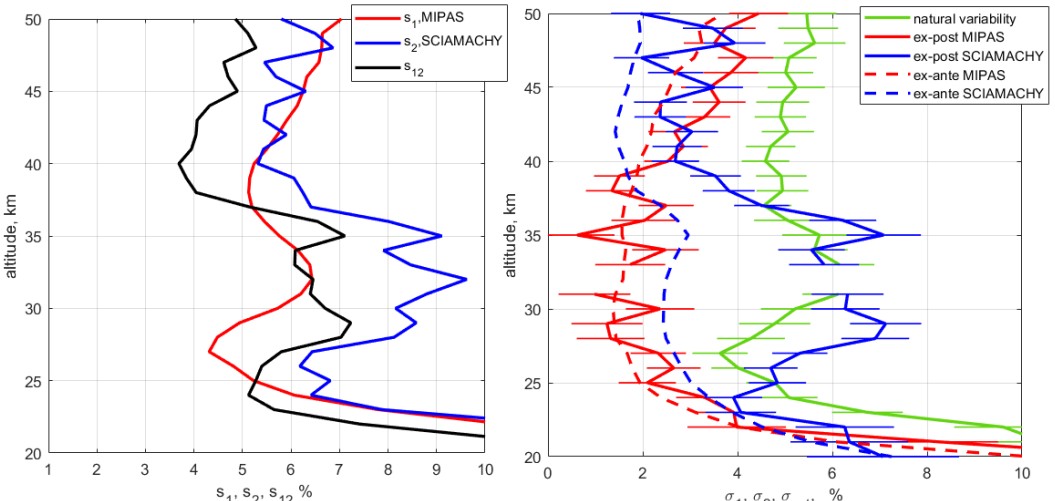

**Figure 4. Application of Fioletov's method to MIPAS and SCIAMACHY ozone datasets in 2007. Left: sample standard deviations $s_1$ and $s_2$ in collocated pairs, and the standard deviation of differences $s_{12}$. Right: a-posteriori uncertainty estimates $\hat{\sigma}_1$ and $\hat{\sigma}_2$, and the estimate of natural variability $\hat{\sigma}_{nat}$ with 1-sigma uncertainties (solid lines). Ex-ante uncertainty estimates are shown with dashed lines.**

As a general note for this and subsequent illustrations, the ex-ante random uncertainties for some instruments (see Table 1 for details) are due to measurement noise. This is a dominating source of random error, however not the only one (von Clarmann et al., 2020). As a result, a posteriori random uncertainty estimates are expected to be slightly larger.

The best performance of the Fioletov's method is expected for datasets with dense sampling and similar random uncertainty estimates. The data should be selected in the regions of low variability. As mentioned above, Fioletov's method requires a large number of collocated profiles to yield reliable estimates of a posteriori uncertainties. The application of this method to solar occultation data by ACE-FTS and SAGE III/ISS is illustrated in Figure 5. The same collocation criteria are used, but the number of collocated profiles is significantly smaller than for MIPAS and SCIAMACHY, even though more years of data are used. For MLS and ACE-FTS, the number of collocations is 741 for years 2018-2019 and 1471 for years 2018-2022. The a posteriori uncertainty estimates from Fioletov's method have substantial error bars (Figure 5 a, b). In application to ACE-FTS and SAGE III/ISS, only 19 collocated profiles are found, so the resulting uncertainty estimates have huge error bars (Figure 5 c).



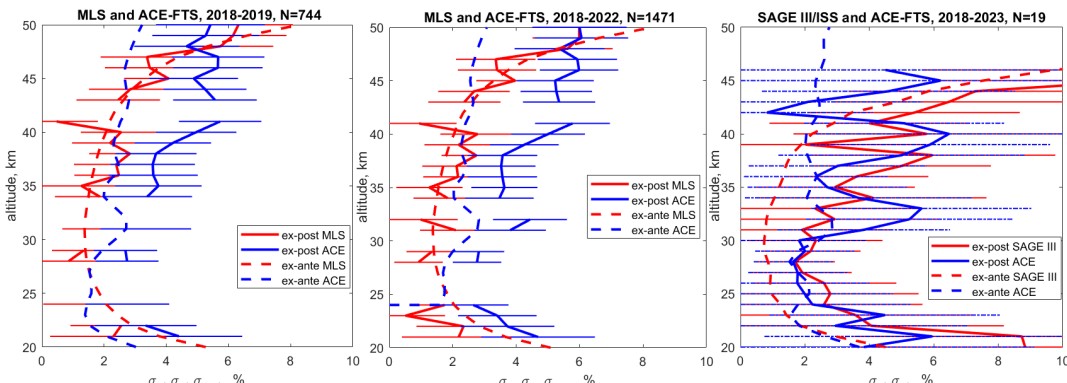

**Figure 5. Ex-ante and ex-post uncertainty estimates from Fioletov's method for MLS and ACE-FTS in years 2018-2019 (a) and 2018-2022(b), and SAGE III/ISS and ACE-FTS in years 2018-2023. The number of collocated profiles is indicated in the panel titles.**

This method is applicable to both individual-profile and tomographic retrievals, as the data in collocated pair can be considered

as uncorrelated. As a general note, the fully proper way of finding co-locations with a tomographic retrieval would be not consider profiles the same way as for individual profile retrievals. Since the along-track dimension is part of the retrieval, interpolation should be done in this dimension to the co-location point.

### 3.4.2    A differential method: comparisons of natural variability patterns

Sofieva et al. (2014) proposed a simple method for detecting flaws and/or checking of consistency of random uncertainty estimates. The authors called it a "differential method". Let us consider, for example, two datasets selected in a region of small and slowly changing natural variability. A large sample size is assumed. If the random uncertainty estimates for both datasets are correct, then the difference in sample variance $s_1^2 - s_2^2$ will be equal to the difference in precision estimates $\sigma_1^2 - \sigma_2^2$. The term $\sigma_{nat}^2$ from Eq. (3) cancels out because it is assumed to be the same for both samples. The estimates of the sample

variance, $s_i^2$, provide the upper limit for experimental estimates of measurement precision, as $s_i^2 > \sigma_i^2$.

A simple comparison of sample variance $s_i^2$ with the random uncertainty estimate $\sigma_i^2$ enables the detection of overestimated random uncertainties, if the relation $s_i^2 > \sigma_i^2$ is violated. Through such a comparison, Sofieva et al. (2024) found overestimated random uncertainties for the GOMOS ozone profiles using very dim stars (further investigation by the instrument experts detected the flaw with accounting instrumental dark charge noise).

If one of the datasets has realistic precision estimates (for example, from well calibrated instruments (so-called Fiducial Reference Measurements), or those estimates are validated by other methods), then application of the differential method is straightforward.

If there are several datasets with unvalidated (or not completely validated) uncertainty estimates, one can consider confronting natural variability estimates $\hat{\sigma}_{nat}^2 = s_i^2 - \sigma_i^2$. Since the natural variability estimates from various datasets should agree within

uncertainty intervals, strong deviations from the majority estimates can potentially indicate flaws in error estimation.

For example, Sofieva et al. (2024) compared estimates of natural variability in the tropics from GOMOS data using different stars and found consistent positive values of $\hat{\sigma}_{nat}^2$ for bright stars ($\chi_{norm}^2 \approx 1$ and application of the structure function method



also suggested that the random uncertainties are realistic for bright stars). However, for very dim stars, negative values of $\hat{\sigma}^2_{nat}$ have been detected, which, together with $\chi^2_{norm} < 1$, pointed to overestimated random uncertainties.

In this paper, we illustrate this differential method by considering sample variance and uncertainty estimates from several limb and occultation instruments. The measurements are selected in the tropics, 20°S-20°N, in three periods, 2002–2004 (first

column of Figure 6), 2006–2008 (2nd column) and 2018-2020 (3rd and 4th column of Figure 6), for the limb instruments operating in these periods. The upper panels of Figure 6 show the sample standard deviation (solid lines) and the mean random uncertainty estimates (dashed lines). The lower panels show the estimates of natural variability $\hat{\sigma}_{nat}$ with associated uncertainties indicated by error bars. For GOMOS, occultations of the 30 brightest stars are used for the analysis, in order to make the GOMOS dataset more homogeneous and to avoid data with overestimated uncertainties. SAGE II and SAGE III/ISS

ozone profiles were smoothed down to 2 km vertical resolution, for compatibility with other datasets.

 As observed in Figure 6, for all datasets except for OMPS USask, $s > \sigma$ as expected. In case of OMPS USask, the mean uncertainty $\sigma$ exceeds the sample standard deviation $s$ at several altitudes, which indicates an overestimation of the random uncertainty component. The overestimation is caused by a bug in the V1.3.0 product and will be fixed in a future version. The profiles of natural variability $\hat{\sigma}_{nat}$ obtained from GOMOS, MIPAS, OSIRIS, ACE-FTS, SAGE II, SAGE III/ISS , OMPS

UBr and OMPS NASA are very close to each other, and their uncertainty intervals overlap. For SCIAMACHY, the pattern of natural variability is also similar, but the increased sample variance at 25–35 km is not explained by its random uncertainty estimates; this suggests a slight underestimation of random error component at these altitudes. For MLS and SAGE III/ISS, there is a very good agreement with other datasets below 40-45 km; above 45 km, the random uncertainty estimates grow fast with altitude, which results in somewhat smaller estimates of $\hat{\sigma}_{nat}$ compared to other datasets. This indicates an overestimation

of MLS and SAGE III/ISS random uncertainties above 40-45 km (see also the explanation in Sect. 2.4).

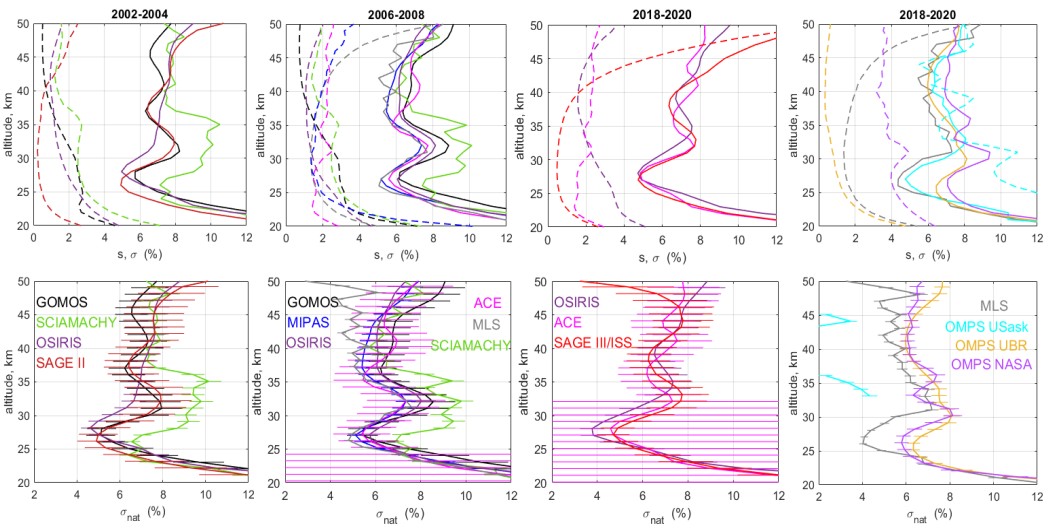

**Figure 6. Top panels: Sample standard deviation $s$ (solid lines) and the mean uncertainty estimates $\sigma$ (dashed lines) in the tropical stratosphere (20°S-20°N) in 2002-2004 (1st column), 2006-2008 (2nd column) and 2018-2020 (3rd and 4th column). Bottom panels: the estimates of the natural variability $\hat{\sigma}_{nat} = \sqrt{s^2 - \sigma^2}$ with its uncertainty (1 std) in the tropics for the same periods. Colors are**

**specified in the bottom panels. For GOMOS, occultations of 30 brightest stars are selected for the analysis.**

Successful application of this method implies the following conditions:





(a) Natural variability should be the same for both samples.

(b) Natural variability should not be large compared to the precision estimates, otherwise the sample variance estimates will have large uncertainty. This condition of small natural variability is satisfied for ozone in the tropical stratosphere and in the summer stratosphere at other latitudes.

(c) Measurements in each sample should have similar precision.

The method can also be applied to the data retrieved with the tomographic approach, if the selected region is sufficiently large (exceeding the horizontal correlation length).

If the natural variability is known from an external source (for example, estimated from the measurements with realistic uncertainty estimates or from a model with the correct variability), a posteriori (ex-post in von Clarmann et al., (2020) terminology) uncertainties can be estimated as $\hat{\sigma}_{ex-post}^2 = s^2 - \sigma_{nat}^2$, where $s^2$ is the sample variance in a set of measurements and $\sigma_{nat}^2$ is the estimated of the natural variability. The use of the modelled data in validation of random uncertainty estimates is also discussed in Sect 3.5 of this paper.

### 3.4.3    Triple collocation methods

#### 3.4.3.1    Stoffelen's method

The idea of using the collocated measurements from three (or more) systems for data calibration and validation of uncertainties was proposed by Stoffelen (1998). In his formulation, it is supposed that three measurement systems X, Y, Z provide collocated measurements of the same quantity $t$. Let system X be the reference system with respect to which systems Y and Z are to be

calibrated. Suppose also that linear calibration (simple scaling) is sufficient for the whole range of values under consideration, and that the reference system X is free of bias. Then the measurements can be written as

$$\begin{aligned}
x &= t + \varepsilon_x \\
y &= c_y(t + \varepsilon_y), \\
z &= c_z(t + \varepsilon_z)
\end{aligned} \qquad (11)$$

where $c_y$ and $c_z$ are scaling factors and $\varepsilon_x$, $\varepsilon_y$, $\varepsilon_z$ are random errors in each measurement sample. The random error components are assumed to be unbiased and not correlated with each other and with the parameter $t$. The calibration

coefficients can be derived from covariances

$$\begin{aligned}
c_y &= \mathrm{cov}(y,z)/\mathrm{cov}(x,z) \\
c_z &= \mathrm{cov}(y,z)/\mathrm{cov}(x,y)
\end{aligned} \qquad , \qquad (12)$$

where $\mathrm{cov}(\cdot,\cdot)$ denotes covariance. These coefficients allow creating the calibrated data $y^* = c_y^{-1}y$, $z^* = c_z^{-1}z$. Then the natural variability of the parameter $t$ can be estimated as e.g. $\sigma_t^2 = \mathrm{cov}(x, y^*)$ and uncertainty variances as

$$\begin{aligned}
\sigma_x^2 &= \mathrm{var}(x) - \sigma_t^2 \\
\sigma_y^2 &= \mathrm{var}(y^*) - \sigma_t^2 . \\
\sigma_z^2 &= \mathrm{var}(z^*) - \sigma_t^2
\end{aligned} \qquad (13)$$

The main assumptions of the method are: (a) the measurements are a linear function of the true signal with additive zero-mean random measurement noise; (b) measurement errors and true signal are stationary, and they are independent; (c) measurement errors are independent, and (d) the measurements are perfectly collocated, i.e. mismatch uncertainty is zero. The assumptions (b-d) are similar to other methods described above. The application of this method to validation of random uncertainties of tropospheric ozone from nadir instruments can be found in Hubert et al. (2021). In case of limb satellite observations, the



requirement of triple collocation reduces dramatically (by an order of magnitude) the length of the sample; this results in larger uncertainties of estimated parameters.

Another variant of the triple collocation method is described in the following subsection.

### 3.4.3.2    Von Clarmann's method

Von Clarmann proposed a method for random uncertainty validation, which used 3 sets of measurements with pairwise collocations (Laeng and Von Clarmann, 2021). This method takes into account the small-scale natural variability, which is estimated using a high-resolution chemistry-transport model data. Let us denote by $s_{ij}^2$ the sample variance of differences in collocated datasets $i$ and $j$, $v_{ij}^2$ natural variability (mismatch) variance in the collocated datasets $i$ and $j$, and $\sigma_i$ ex-ante uncertainty estimates, and assume that the true random uncertainties are $\sigma_{true,i}^2 = c_i \sigma_i^2$, $i = 1, 2, 3$. Then the expression for the sample variance in the collocated pairs results in the following system for determination of correction factors $c_i$

$$
\begin{aligned}
c_1\sigma_1^2 + c_2\sigma_2^2 + v_{12}^2 &= s_{12}^2 \\
c_1\sigma_1^2 + c_3\sigma_3^2 + v_{13}^2 &= s_{13}^2 \\
c_2\sigma_2^2 + c_3\sigma_3^2 + v_{23}^2 &= s_{23}^2
\end{aligned}
\tag{14}
$$

If $v_{ij}^2$ are known, the solution of the linear system (14) is

$$
\begin{aligned}
c_2 &= \frac{1}{2\sigma_1^2}\left[ (s_{12}^2 - v_{12}^2) + (s_{13}^2 - v_{13}^2) - (s_{23}^2 - v_{23}^2) \right] \\
c_2 &= \frac{1}{2\sigma_2^2}\left[ (s_{12}^2 - v_{12}^2) + (s_{23}^2 - v_{23}^2) - (s_{13}^2 - v_{13}^2) \right] \\
c_3 &= \frac{1}{2\sigma_3^2}\left[ (s_{13}^2 - v_{13}^2) + (s_{23}^2 - v_{23}^2) - (s_{12}^2 - v_{12}^2) \right]
\end{aligned}
\tag{15}
$$

The Eqs .(14) and (15) are written in terms of correction factors, as it is presented in the original report   however, they can be also presented in  terms of ex-ante and ex-post uncertainties.   As for Fioletov's method, Eq.(15)  does not require positivity of $c_i$ (solution of the linear system), which may result in unphysical negative estimates of random error variance. Since the estimates by Eq. (15) are the linear combinations of sample variances, the accurate estimate require many collocated data, similarly to the Fioletov method.

For illustration, the method was applied to MIPAS, MLS, and SCIAMACHY data in 2007 at 20°S–20°N, where 3236 triple collocations (with time difference < 4h and spatial separation < 300 km) are found. The small-scale variability estimates were obtained from BASCOE model data field down-sampled to typical horizontal resolution along line of sight of limb instrument data (Laeng et al., 2022).





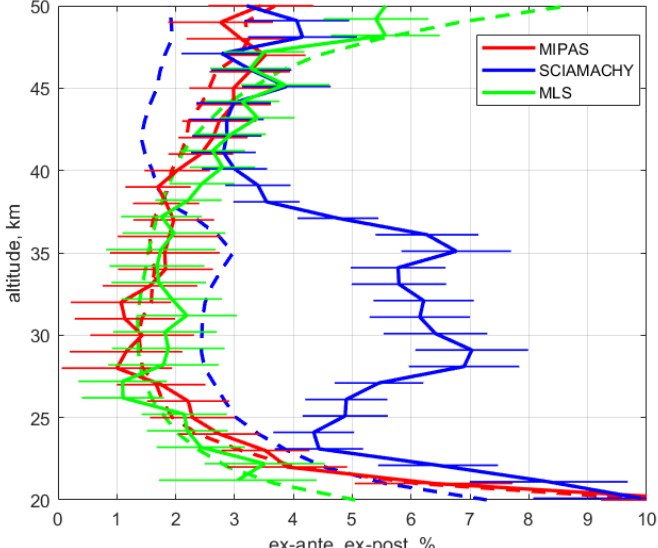

**Figure 7. Dashed lines: ex-ante uncertainties; solid lines with error bars: ex-post random uncertainties estimated by the von Clarmann method.**

5    The use of the small-scale variability from the ozone field generated by an advanced chemistry-transport model with a high horizontal resolution seems to be correct, as this mismatched variability is caused by dynamics, and it is characterized by the statistical characteristic (variance). Although some small-scale processes may not be resolved in a model, this does not influence the $v_{ij}$ estimates, as the effective horizontal resolution of the limb measurements along the line of sight is ~300 km.

The results of application of the von Clarmann's method (Figure 7) agree very well with other methods presented in our paper.

### 3.5    Using CTM simulation in validation of uncertainties

Modern chemistry-transport models have high horizontal and vertical resolution, and a majority of them use meteorological reanalyses in the advection schemes. They show good agreement with the observational data, therefore it is attractive to use the information the models provide in validation of uncertainties. For example, the modelled field can be used for

characterization of differences due to the co-location mismatch, i.e. differences in spatio-temporal sampling and smoothing of the variable and inhomogeneous ozone field. Such an approach has been applied in several studies ( e.g., Sheese et al., 2021; Verhoelst et al., 2015).

The model estimates of small-scale natural variability are used also in von Clarmann's method. Potentially, analogous characterization would also improve the Fioletov's method.

Sofieva et al. (2022) used ozone data, which are simulated with the chemistry-transport model SILAM adjusted to MLS for a posteriori random uncertainty estimates by the differential method. For each instrument and each month, the authors evaluated sample variance in 10° latitude zones from experimental data and the SILAM-adjusted field, which is sub-sampled at measurements locations. The model sample variance provides the estimates of natural variability. Then a posteriori uncertainties were estimated as $\hat{\sigma}^2_{ex-post} = s^2 - \sigma^2_{nat}$, where $s^2$ is the sample variance in a set of measurements and $\sigma^2_{nat}$ is the

estimate of the natural variability. Figure 8 illustrates ex-ante and ex-post uncertainties for GOMOS, MIPAS, SCIAMACHY, OSIRIS and MLS using the data in September 2007. These estimates agree very well with those obtained by the Fioletov's



method (Figure 4, right). The approach of Sofieva et al. (2022) allows selecting sufficiently large data samples in a relatively short time period ; the authors applied their method to the adjustment of random uncertainties for each month.

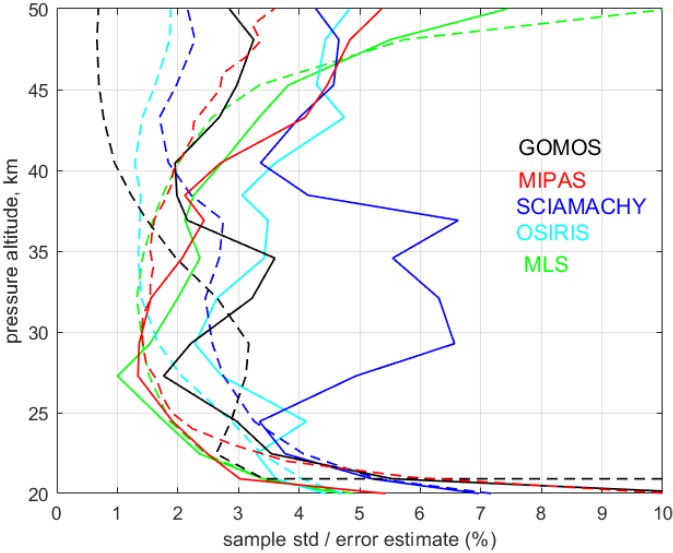

**Figure 8. Ex-ante (dashed lines) and ex-post (solid lines) random uncertainty estimates for September 2008, based on the method**
**described in Sofieva et al. (2022)**

### 3.6 Notes on validation with fiducial reference measurements

If a dataset with well known (or validated and concluded to be realistic) uncertainty estimates is available, then the validation of uncertainties of a second dataset using Eq. (4) or the differential method is straightforward. Such an approach is usually
used in validation of satellite measurements with ground-based data. When exploiting Eq. (4), it is advantageous to characterize/simulate $\sigma_{0,\text{var}}^2$. Such approach was explored for the validation of satellite total ozone column data by ground-based measurements (Verhoelst et al., 2015).

For ozone profiles, ozonesondes are usually used for validation of satellite data (evaluation of biases and drifts). However, according to (Tarasick et al., 2021) the characterization of ozonesonde uncertainties is even more complicated than for satellite
data, and the uncertainties are not constant but varying in the range 5-10% (sometimes even up to 20%). Together with a limited number of tight collocations, these features impose limitations in validation of satellite random uncertainties using ozonesonde data.

### 3.7 On using the Markov Chain Monte Carlo method

The random uncertainties reported by retrieval algorithms are usually estimated via propagation of instrumental noise and
other random uncertainties through the inversion algorithm. Markov chain Monte Carlo (MCMC) method can be used to produce a robust estimate of the probability distribution of a retrieved quantity that is nonlinearly related to the measurements and that has non-Gaussian error statistics. A methodology for validating the traditional error characterization by applying the MCMC technique can be found in e.g. Tamminen (2004). This paper shows the application of MCMC method to GOMOS data. The MCMC technique is suitable for studying uncertainties of retrieved parameters and it enables analyzing the error
structure also in a nonlinear case (and thus validating the standard Gaussian characterization). The advantage of the sampling



based MCMC method is also that it allows implementing non-Gaussian measurement and modelling error characterization as well as using non-Gaussian prior information. While the MCMC method cannot provide information about missing or overestimated uncertainties directly, the method is often implemented so that unknown uncertainties are parametrized and included, e.g., via hierarchical formulation, allowing these uncertainties to be taken into account. It would be very useful to compare such approaches with the ones presented in this overview paper, in the future.

## 4    Summary

In this paper, we presented methods for random uncertainty validation, which were illustrated using ozone profiles retrieved from measurements by satellite instruments in the limb-viewing geometry. These methods rely on deriving a-posteriori random uncertainties using statistical analyses of collocated data samples. Advantages and limitations of each method are discussed, as well as accuracy of a-posteriori random uncertainty estimates.

As a general requirement for all methods, the data samples should be selected in regions of small and slowly changing natural variability. Otherwise, if the natural variability exceeds significantly random uncertainties, this prevents the computation of reliable a-posteriori estimates of random uncertainties. The methods for random uncertainty validation are divided into two groups depending on what kind of data are used: (1) from the same instrument and (2) from different instruments.

Practical examples for validation of random uncertainty with the discussion of advantages and limitations of each method are provided in this study. It is shown that, for instruments with dense sampling, such as MIPAS and MLS, several methods can be applied, for example those based on self-collocations or collocations with other datasets. For datasets that are obtained with tomographic retrievals, the Fioletov's, von Clarmann and differential methods can be applied. For instruments with coarse sampling, such as GOMOS, ACE-FTS or SAGE II-III, the differential method is the most appropriate. It has been shown previously and confirmed in this study that simulations with high-quality and high-resolution chemistry-transport models are useful in validation of reported random uncertainties: the model simulations can be used for estimation of small-scale natural variability.

## 5    Acknowledgements

We acknowledge the scientific guidance (and sponsorship) of the World Climate Research Programme to motivate this work, coordinated in the framework of APARC TUNER activity. The International Space Science Institute (ISSI) has funded two International Team meetings in Berne at their venue. Part of this work was funded by ESA under contract no. 4000128645/19/I-DT. VFS and JT thank thanks the Academy of Finland (Centre of Excellence of Inverse Modelling and Imaging; decision 353082). AL performed her contribution within the ESA VACUUM project. The authors thank the Canadian Space Agency and the Swedish National Space Board for their long-term support of the Odin mission and the OSIRIS instrument. The work at the University of Bremen was funded in parts by ESA and University and State of Bremen. We gratefully acknowledge the computing time granted by the Resource Allocation Board and provided for the supercomputers Lise and Emmy at NHR@ZIB and NHR@Göttingen as part of the NHR infrastructure.

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
