# Peer review of "Methods for validation of random uncertainty estimates and their applications to ozone profiles from limb-viewing satellite instruments"

_EGUsphere, 2025_

## Referee Comment (RC2)

Review of the paper "Methods for validation of random uncertainty estimates and their applications to ozone profiles from limb-viewing satellite instruments" by V. Sofieva et al.

**General comments**

This paper provides an overview of methods for validating the random-error components of uncertainties in remote-sensing products. While these methods have been described previously in the literature, the authors bring them together here, apply them to various limb-sounding observations, and compare their performance for ozone profiles. The topic is relevant for data providers and data users, and the manuscript is generally well written. Some harmonisation of terminology would be beneficial, however, and the effect of retrieval constraints on the validation of the random error could be better addressed (see specific comments below).

Overall, I believe the manuscript can be published in *AMT* after the authors address the minor issues raised in this review.
* * *
**Specific comments**

1. **Consistency of terminology**
   o for measurement noise error
     The propagation of instrument noise into the retrieval is referred to by several different names throughout the manuscript—for example, *measurement noise error* (pag. 3, line 29), *ex-ante estimate for instrumental noise* (pag. 9, line 12), and *propagation of instrumental noise* (pag. 1, line 38), while *measurement noise* at pag.4 (lines 9 and 10) is the error in the measurement domain. I recommend reviewing the entire manuscript and adopting a single, consistent term.
   o in the description of all instrument and in particular in Section 2.5.
     The presentation of the different cases would benefit from improved harmonisation. Please highlight clearly which aspects are common among the cases and which aspects differ.
2. **Clarity regarding the handling of a priori information**
   Statements related to the use of a priori constraints could be better explained.
   o Page 6, lines 23–25: "If the retrieval is performed with the Bayesian maximum a posteriori estimates, a data correlation can also arise due to the usage of a priori information. These aspects should be taken into account when validating uncertainties."
     It is not evident whether these aspects *have* been taken into account in the analyses presented, apart from the MLS case, for which it is said that "it overestimates the scatter in the geophysical products in cases where the a priori information and other regularisation constraints contribute significantly to the results," which implies that smoothing error *is* included in the random error. In the OMPS-LP USask case it is specified that smoothing error is not included in the random error; does this mean that smoothing error is included for all other instruments/dataset?
     A clearer and more systematic description of how a priori information and smoothing error are treated for each instrument, also reporting the equation for the random error, would help.
   o Page 7, lines 27–29: The manuscript states that it is important for this type of analysis that all profiles have similar vertical resolution. Could you better clarify the

rationale?

Is the requirement related to the relative strengths of the constraints used in each retrieval? While vertical resolution certainly depends on the constraint strength, it also depends on the measurement vertical sensitivity and the chosen retrieval grid.

o   In general a short description of what the validation of random error has to consider in presence of constraints in the inversion procedure and with the inclusion of the smoothing error in the random error would be valuable, also because it would help readers understand how the presented validation techniques could be extended, for example, to nadir observations

**Minor corrections**

o   Pag. 6, line 10: an estimate OF THE
o   Pag.5, line 25: please remove 'is provided'
o   Caption of Fig. 3:
   1. vurves -> curves
   2. uncertainty estimate AND its standard deviation
o   Eq. 15: First $c_2$ -> $c_1$

---

## Author Comment (AC1)

Dear Reviewer,

Thank you very much for your comments on our paper. We took your comments into account in the revised version of the manuscript. Please find below our detailed replies (black font) on your comments (blue font).

Reviewer#1 comments:

The paper discusses various methods for assessing the actual random error of satellite-derived profiles of atmospheric data, comparing these with the reported random error, and showing results of various methods of assessment.

I have no objection to the content of the paper, or the analysis, but I offer some suggestions for improvements and clarifications which the authors may wish to consider.

1) Firstly, There ought to be an initial statement of the meaning of 'random uncertainty' as used in the title of this paper. I think it may be close to the lab definition at the start of section 3.2 but it should be at the start, along with some discussion of various alternative interpretations that have also been used (e.g. those briefly listed in Table 1).

In the revised version, we added a short definition of random uncertainty.

2) I can think of two further methods which could also at least be mentioned, if not applied.
a) (in addition to methods considered in 3.3) For continuous limb-scanning instruments retrieving profiles every few hundred km along the orbit, one could compare each profile with a profile interpolated from the profiles immediately before and afterwards along the orbit. While this still has some component of natural variability, that would be reduced by the linear interpolation. This has an advantage over the method of using orbital intersections since the time gap is smaller and all three profiles are likely to be measured with the same day or night illumination. However, this would not work for tomographic retrievals.

Thank you for the interesting idea. However, the interpolated profile cannot be considered as "measurement": it can be different from reality, and its uncertainty will be correlated with uncertainties of data used for interpolation. In terms of validation of uncertainties, interpolation will not bring additional information.

For the limb instruments with dense sampling, one can consider comparison of pairs of consecutive measurements along the orbit. However, the spatial separation of a few hundred km is not sufficiently small, so that the variance of differences contain the typical patterns of small-scale variability.

b) (in addition to methods considered in 3.4) Measurement of variation about the zonal mean. This seems to be partly covered in 3.4.2 but I was thinking of much narrower latitude bands. This particularly suits solar-occultation instruments which typically make 14 measurements in each of two very tightly-constrained latitude bands every 24 hours, and also tomographic retrievals since any along-orbit correlation is likely to be negligible over half an orbit. One could then dispense

with any time (apart from within the same day) or longitudinal constraint on matching - hence more comparisons with polar-orbiting instruments - and the only additional information required is \sigma_nat on c.15deg longitude scale which is, obviously, the same for all instruments and, in the summer stratosphere, quite possibly negligible

The latitude band can be narrower than that used for illustration in Section 3.4.2. However, evaluation of sample variance requires large number of measurements. For the particular case of solar occultation, it requires combining the data from several days /latitudes.

For measurements along the orbit, the natural variability is so large that any estimates of random uncertainties are not reliable (see requirements in Sect 3.4.2)

3) Although the various methods that are discussed are applied to different instruments, there is no summary table or plot comparing the results from the different methods applied, eg, to just one instrument, so that the methods can be directly compared.

In the revised version, we included a figure comparing ex-post uncertainties for MIPAS at 20°S-20°N in 2007 estimated by different methods and the corresponding discussion.

Minor points/typographical corrections:

Section 2 - it would be helpful in each subsection to have just an initial sentence describing the type of instrument/observation.

In the revised version, we added a short description of type of instruments and also the main principle of ozone profile retrievals. The revised Table 1 also contains information about the retrieval method.

Generally, use 'en' dashes ($--$) to indicate a range of numbers rather than hyphens (eg Figure captions, P11 L20, P13 L4-5 L18, P17 L15).

Corrected

P5 L9 (&L17): A large chi-squared value seems more likely to indicate the presence of residual spectral features, eg systematic errors in the forward model, than correctness of the assumed random error.

This sentence begins with "If the theoretical model describes the experimental data correctly,…" i.e., this is the statement for the case when there are no systematic (and big) misfit of spectra.

P5 L20: 'em' dashes are required here ($---$ in LaTeX),

P7 L19/20: 'which represent ... is different': sigma^2_0,nat is treated as both plural and singular in this sentence.

Corrected

P8 L9: Note that such collocated measurements necessarily involve comparing ascending and descending nodes of the orbit, so likely to involve different day/night conditions.

In the revised version, we added this note.

P8 Eq (5): presumably D(\rho) depends differently on the magnitude of each coordinate of \rho (and in any case some scaling is required to convert between the time and space coordinates).

Yes, $D(\boldsymbol{\rho})$ is usually anisotropic (illustrations can be found in Sofieva et al., 2021).

P9 L11: $S_{12}$ (upper case here, lower case elsewhere)

P9 Fig 15 caption: '20011' should presumably be '2011'.

P10 L16: "true" - initial pair of double-quote marks show as ",,"

Corrected

P10 L22: "not dense" - I suggest "sparse"

Changed as suggested.

P10 L32: Here it seems that "a-posteriori" and "ex-post" mean the same thing but elsewhere both are used individually so it is less clear that their meanings are the same. Also "a posteriori" is sometimes hyphenated, sometimes not (P16 L21)

In the revised version, "a posteriori" is changed to "ex-post" everywhere except its first definitions and introduction of von Clarmann terminology.

P11 L2: Since it is a direct part of the sentence, I would suggest "von Clarmann et al (2020)" rather than "(von Clarmann et al., 2020)" (also P17 L14)

Corrected

P13 Fig 25: I was initially impressed with the consistency of the \sigma_nat values shown in the lower plots, but then I realised that these are very similar to the sample SDs shown in the upper plot, somewhat contradicting condition (b) mentioned on P14 L2.

We agree. The smallest natural variability is in the tropics, but the random uncertainty estimates for limb-instruments are usually smaller. In the revised version, we added this note.

P14 L23: pedantically it should perhaps be noted that \epsilon_y,z refer to random errors scaled to x rather than associated with the original measurements (to me it seems more natural to have eg y = c_y t + e_y)

We follow the original formulation of Stoffelen (1998).

P12 L11: I may have missed it, but what is $\sigma^2_0,var$ ?

It should be $\sigma_{0,nat}^2$ . We corrected this.

---

## Author Comment (AC2)

Dear Reviewer,

Thank you very much for your comments on our paper. We took your comments into account in the revised version of the manuscript. Please find below our detailed replies (black font) on your comments (blue font).

Reviewer#2 comments:

This paper provides an overview of methods for validating the random-error components of uncertainties in remote-sensing products. While these methods have been described previously in the literature, the authors bring them together here, apply them to various limb-sounding observations, and compare their performance for ozone profiles. The topic is relevant for data providers and data users, and the manuscript is generally well written. Some harmonisation of terminology would be beneficial, however, and the effect of retrieval constraints on the validation of the random error could be better addressed (see specific comments below).

**Specific comments**

1. *Consistency of terminology*

- for measurement noise error
  The propagation of instrument noise into the retrieval is referred to by several different names throughout the manuscript—for example, *measurement noise error* (pag. 3, line 29), *ex-ante estimate for instrumental noise* (pag. 9, line 12), and *propagation of instrumental noise* (pag. 1, line 38), while *measurement noise* at pag.4 (lines 9 and 10) is the error in the measurement domain. I recommend reviewing the entire manuscript and adopting a single, consistent term.

We revised using the term "noise". It is now everywhere either "noise" (at the instrument level) or "propagated measurement noise" (for retrieved ozone profiles).

- in the description of all instrument and in particular in Section 2.5. The presentation of the different cases would benefit from improved harmonisation. Please highlight clearly which aspects are common among the cases and which aspects differ.

In the revised version, we added a short description of type of instruments and also the main principle of ozone profile retrievals and error propagation, for all datasets. The revised Table 1 also contains information about the retrieval method.

2. **Clarity regarding the handling of a priori information**

Statements related to the use of a priori constraints could be better explained.
- Page 6, lines 23–25: "If the retrieval is performed with the Bayesian maximum a posteriori estimates, a data correlation can also arise due to the usage of a priori information. These aspects should be taken into account when validating uncertainties." It is not evident whether these aspects *have* been taken into account in the analyses presented, apart from the MLS case, for which it is said that "it overestimates the scatter in the geophysical products in cases where the a priori information and other regularisation constraints contribute significantly to

the results," which implies that smoothing error *is* included in the random error. In the OMPS-LP USask case it is specified that smoothing error is not included in the random error; does this mean that smoothing error is included for all other instruments/dataset? A clearer and more systematic description of how a priori information and smoothing error are treated for each instrument, also reporting the equation for the random error, would help.

For all datasets used in the paper, only regularization is applied in the retrievals for majority of datasets. Since the vertical resolution is approximately the same for all considered datasets, "smoothing error" also approximately the same. The smoothing error has both systematic and random components. For limb instruments, the random component smoothing error is usually not estimated: to estimate smoothing error properly, information about high-vertical resolution variability is needed (and this information is not available). Since the vertical resolution of the considered ozone profiles is approximately the same for different datasets, ex-ante random uncertainties are characterized consistently. In the revised version, we presented more details on retrievals (see also above).

- Page 7, lines 27–29: The manuscript states that it is important for this type of analysis that all profiles have similar vertical resolution. Could you better clarify the rationale? Is the requirement related to the relative strengths of the constraints used in each retrieval? While vertical resolution certainly depends on the constraint strength, it also depends on the measurement vertical sensitivity and the chosen retrieval grid.

Since estimates presented in our paper are based on the statistics of differences, the profiles should be compatible, i.e., have similar vertical resolution. This is required in order to be able to neglect vertical smoothing difference errors. In the revised version, we added this note. Natural variability also depends on the vertical resolution. For limb instruments, the vertical resolution depends mainly on constraint strength.

- In general a short description of what the validation of random error has to consider in presence of constraints in the inversion procedure and with the inclusion of the smoothing error in the random error would be valuable, also because it would help readers understand how the presented validation techniques could be extended, for example, to nadir observations

We extended the summary with a short discussion:

"The methods presented in this overview can be also applied to other measurements. In particular, the structure function method has been already successfully applied to total column measurements by TROPOMI in Sofieva et al. (2021). All methods can be applied also to data with coarse vertical resolution, such as profiles retrieved from nadir-looking instruments. For the application of the methods based on the statistics of differences, the profiles should have a compatible vertical resolution. This might require prior application of harmonization ( see Keppens et al. (2019) for details). Then the validation of random uncertainties can be performed at the vertical scales corresponding to harmonized profiles"

**Minor corrections**
- Pag. 6, line 10: an estimate OF THE
- Pag.7, line 25: please remove 'is provided'

- Caption of Fig. 3:
    - 1. vurves -> curves
    - 2. uncertainty estimate AND its standard deviation
- Eq. 15: First c2 -> c1

Corrected